# Stepwise transmigration of T- and B cells through a perivascular channel in high endothelial venules

Kibaek Choe[1], Jieun Moon[1], Soo Yun Lee[1], Eunjoo Song[1], Ju Hee Back[1], Joo-Hye Song[3], Young-Min Hyun[4], Kenji Uchimura[5,6], Pilhan Kim[1,2]

**High endothelial venules (HEVs) effectively recruit circulating lymphocytes from the blood to lymph nodes. HEVs have endothelial cells (ECs) and perivascular sheaths consisting of fibroblastic reticular cells (FRCs). Yet, post-luminal lymphocyte migration steps are not well elucidated. Herein, we performed intravital imaging to investigate post-luminal T- and B-cell migration in popliteal lymph node, consisting of trans-EC migration, crawling in the perivascular channel (a narrow space between ECs and FRCs) and trans-FRC migration. The post-luminal migration of T cells occurred in a PNAd-dependent manner. Remarkably, we found hot spots for the trans-EC and trans-FRC migration of T- and B cells. Interestingly, T- and B cells preferentially shared trans-FRC migration hot spots but not trans-EC migration hot spots. Furthermore, the trans-FRC T-cell migration was confined to fewer sites than trans-EC T-cell migration, and trans-FRC migration of T- and B cells preferentially occurred at FRCs covered by CD11c+ dendritic cells in HEVs. These results suggest that HEV ECs and FRCs with perivascular DCs delicately regulate T- and B-cell entry into peripheral lymph nodes.**

## Introduction

Lymph nodes constantly recruit and return lymphocytes to and from the blood to facilitate rapid encounters between antigens and rare antigen-specific lymphocytes (1, 2). Circulating lymphocytes in the blood enter the lymph nodes via high endothelial venules (HEVs), the wall of which is composed mainly of two cellular components, endothelial cells (ECs) and fibroblastic reticular cells (FRCs) (2). A current model of lymphocyte transmigration across the HEV wall consists of four distinct steps (2): rolling (initiation of adhesion between lymphocytes and HEV ECs), sticking (firm adhesion to ECs), intraluminal crawling (searching for a suitable exit

site), and trans-endothelial migration (trans-EC migration). After the trans-EC migration, however, lymphocytes must crawl inside the perivascular channel (PVC) (1, 3), a narrow space between ECs and FRCs, and subsequently transmigrate across FRCs to finally arrive at the lymph node parenchyma. Boscacci et al described the intra-PVC crawling of T cells as perivascular trapping around HEVs due to a delay in directed motility and low directional persistence in the perivascular region within 20 μm of the HEV endothelium (4). Park et al reported that B cells remained flatten along the abluminal side of HEVs after trans-EC migration (5). Although efforts to elucidate post–trans-EC migration in HEVs have been made, a clear visualization and molecular mechanism of post–trans-EC migration, including the intra-PVC and trans-FRC migration of T- and B cells in HEVs, is still lacking.

L-selectins expressed on lymphocytes are well-known adhesion molecules that mediate the initiation of lymphocyte rolling along the luminal side of HEVs by binding to their ligands expressed on the HEV endothelium (1, 6). Interestingly, L-selectin ligands are expressed not only on the luminal side but also on the abluminal side of the HEV endothelium (7). Peripheral node addressins (PNAds), the main L-selectin ligands, require carbohydrate sulfation for optimal L-selectin binding. The sulfation is catalysed by GlcNAc-6-O-sulfotransferases (GlcNAc6STs), of which GlcNAc6ST-1 and GlcNAc6ST-2 contribute to PNAd expression in HEVs (8, 9). GlcNAc6ST-1 deficiency leads to lower PNAd expression on the abluminal side of HEVs and reduces the number of lymphocytes entering the lymph node (10). However, the detailed effects of GlcNAc6ST-1 deficiency on the abluminal migration of lymphocytes remain elusive.

During the diapedesis of T cells across HEVs, T cells do not exit through random sites in HEVs but rather through discrete sites, called "exit ramps" (11). Many studies have been performed to understand the hot spots of trans-EC and trans-pericyte neutrophil migration in inflamed tissue (12, 13, 14, 15, 16). However, the hot spots of T- and B cells in lymph nodes remain poorly understood despite the fact that the ECs and FRCs of HEVs are substantially different

[1]Graduate School of Nanoscience and Technology, Korea Advanced Institute of Science and Technology, Daejeon, Republic of Korea  [2]Graduate School of Medical Science and Engineering, Korea Advanced Institute of Science and Technology, Daejeon, Republic of Korea  [3]Center for Vascular Research, Institute for Basic Science, Daejeon, Republic of Korea  [4]Department of Anatomy and Brain Korea 21 PLUS Project for Medical Science, Yonsei University College of Medicine, Seoul, Republic of Korea  [5]Department of Biochemistry, Nagoya University Graduate School of Medicine, Nagoya, Japan  [6]Unité de Glycobiologie Structurale et Fonctionnelle, UMR 8576 CNRS, Université de Lille, Villeneuve d'Ascq, France

Correspondence: pilhan.kim@kaist.ac.kr

from the ECs and pericytes of normal venules in nonlymphoid organs ([17]).

Herein, we clearly visualized the multiple steps of post-luminal T- and B-cell migration in popliteal lymph node, including trans-EC migration, intra-PVC crawling, and trans-FRC migration, using intravital confocal microscopy and fluorescent labelling of ECs and FRCs with different colours. Our 3D cell tracking analysis revealed that GlcNAc6ST-1 deficiency led to T- and B cells requiring more time for trans-FRC migration. In addition, PNAd blocking increased the amount of time required for trans-EC and trans-FRC T-cell migration and delayed the passage of T cell in PVC by making the T-cell detour to an exit site. Next, we found the hot spots for T- and B-cell trans-EC and trans-FRC migration. Simultaneously imaging T- and B cells showed that T- and B cells preferentially shared the hot spots for trans-FRC migration but not for trans-EC migration. Interestingly, trans-FRC T-cell migration was confined to fewer sites than trans-EC T-cell migration, and T- and B cells prefer to transmigrate across FRCs covered by CD11c+ DCs in HEVs. These results imply that FRCs delicately regulate the transmigration of T- and B cells across the HEV wall, which could be mediated by perivascular DCs.

# Results

## Intravital imaging of T- and B-cell transmigration across HEVs composed of ECs and FRCs

To clearly visualize the multiple steps involved in post-luminal T-cell migration in HEVs by intravital confocal fluorescence microscopy, we adoptively transferred GFP-expressing T cells (green) and injected fluorescence-labelled anti–ER-TR7 antibody (blue) into an actin-DsRed mouse (red; Fig 1A and B). By acquiring time-lapse Z-stack images, we observed many 3D T-cell tracks in HEV (Video 1). Representative serial images (Figs 1C and S1) clearly show the multiples steps of T-cell migration across HEV composed of ECs and FRCs through the perivascular channel (a narrow space between ECs and FRCs): adhesion to ECs, intraluminal crawling, trans-EC migration, intra-PVC crawling, and trans-FRC migration, finally arriving at the lymph node parenchyma. To compare T-cell and B-cell migration, we also performed the same experiment with B cells (Fig 1C). 3D tracking analysis showed that B cells required more time for trans-EC migration (3.0 ± 2.2 min) and trans-FRC migration (1.8 ± 0.9 min) than T cells (trans-EC time, 1.5 ± 0.9 min; trans-FRC time, 1.5 ± 0.9 min; Fig 1D and F). The mean velocity of T cells (5.3 ± 1.7 $\mu$m/min) was significantly higher than that of B cells (4.1 ± 1.4 $\mu$m/min) during intra-PVC migration (Fig 1E), whereas the dwell time and total path length in the PVC were not significantly different between T- and B cells (Fig 1H and I). Similar results were obtained when both cells were imaged simultaneously, except that B cells had significant longer dwell time than T cells (Figs 2C–F and S5). Interestingly, more than half of the T- and B cells crawled from 50 to 350 $\mu$m inside the PVC (Fig 1I), which implies that T- and B cells are not stationary but rather actively search for suitable exit sites inside the PVC. In addition, the linear dependence of the path length and dwell time in the PVC (Fig 1J) implies that some T- and B cells leave the PVC more quickly not because of the higher velocity but rather

because of the shorter path length to exit sites. We also measured the velocity of T- and B cells in parenchyma, revealing no significant changes in their velocity at 1-min intervals up to 10 min after trans-FRC migration (Fig S2). The mean velocity of T cells in parenchyma (8.9 ± 2.3 $\mu$m/min) was significantly higher than that of B cells (5.3 ± 1.8 $\mu$m/min; Fig 1G). To investigate the effect of injecting an anti–ER-TR7 antibody into the mouse footpad on T-cell migration in HEVs, we performed the same experiment in the absence of the antibody. The trans-EC and trans-FRC migration times, dwell time, path length, and mean velocity in the PVC were similar between the antibody-injected and noninjected groups, whereas the mean velocity of parenchymal T cells of the antibody-treated group was higher than that of non-injected parenchymal T cells (Fig S3). This result shows that injection of the anti–ER-TR7 antibody does not affect T-cell migration in HEVs but may increase the velocity of T cells in parenchyma, which appears to be related to a previous report that T cells crawl along the FRC network in lymph node parenchyma ([11]).

## Nonredundant role of L-selectin/PNAd interactions in the post-luminal migration of T- and B cells in HEVs

To explore the molecular basis underlying the post-luminal migration of T- and B cells in HEVs, we performed intravital imaging of GlcNAc6ST-1 KO mice that have low PNAd expression on the abluminal side of HEVs ([8], [9], [10]). To simultaneously image T- and B cells, we adoptively transferred DsRed-expressing T cells (red) and GFP-expressing B cells (green) into GlcNAc6ST-1 KO and wild-type mice (Fig 2A and B). The trans-FRC migration times of T- and B cells in KO mice (T, 2.5 ± 1.7 min; B, 3.6 ± 2.2 min) were significantly longer than those in wild-type mice (T, 1.6 ± 1.1 min; B, 1.9 ±1.1 min; Figs 2E, G, and H and S4 and Video 2). In contrast, the trans-EC migration times of T- and B cells were similar between the two groups (Fig 2C). This indicates that GlcNAc6ST-1 was required for efficient T- and B-cell trans-FRC migration, but not for their trans-EC migration. The mean velocities of B cells in the PVC (3.0 ± 0.7 $\mu$m/min) and even in the parenchyma (3.7 ± 0.9 $\mu$m/min) of KO mice were substantially lower than those of wild-type mouse B cells (in PVC, 3.5 ± 0.8 $\mu$m/min; in parenchyma, 4.5 ± 1.3 $\mu$m/min; Fig 2D and F), whereas those of T cells were similar between the two groups. This indicates that GlcNAc6ST-1 was involved in intra-PVC migration as well as even in parenchymal migration for B cells. The fact that B cells were more affected than T cells is consistent with a previous report on T- and B-cell rolling and sticking in GlcNAc6ST-1 KO and GlcNAc6ST-1/2 double KO mice ([9]), which was attributed to the 1.5-fold higher L-selectin expression in T cells than that in B cells ([18]). The dwell times of T- and B cells in the PVCs of KO mice were slightly increased (T, 27 ± 22 min; B, 42 ± 31 min) compared with those of wild-type mice (T, 21 ± 18 min; B, 30 ± 22 min), although statistical significance was not reached (Fig S5A). The path lengths, displacement, and meandering indices (MIs) of PVCs in T- and B cells were similar between the two groups (Fig S5B–E). In addition, there was no difference in the percentage of homing T-cell subsets (central memory, naïve CD4, and naïve CD8) between KO and wild-type mice (Fig S6). These results imply that lower PNAd expression on the abluminal side of HEVs due to GlcNAc6ST-1 deficiency may lead to delay T- and B-cell trans-FRC migration and to decrease the B-cell mean velocities in PVC and parenchyma.

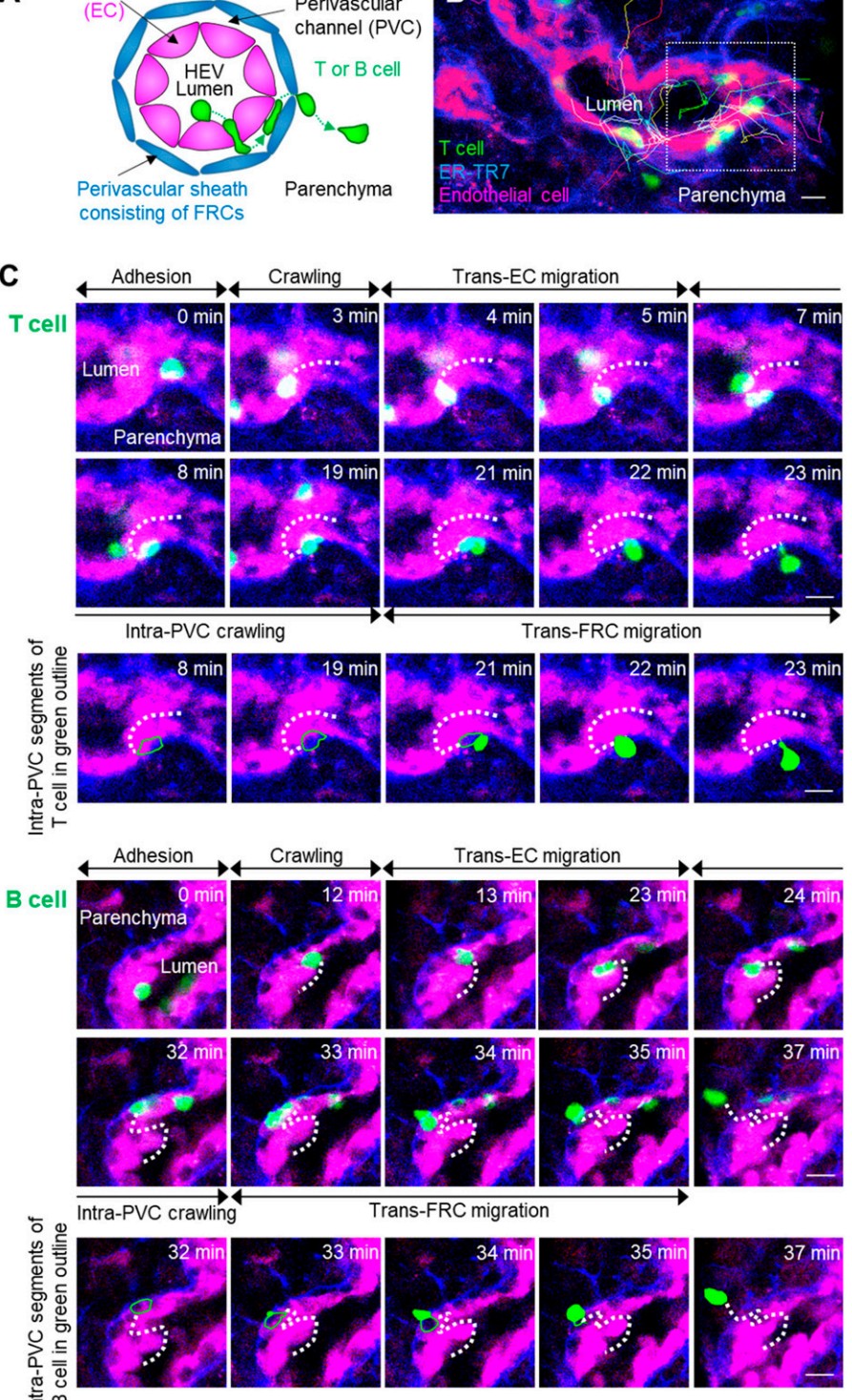

**Figure 1. Intravital imaging of T- and B-cell transmigration across high endothelial venules via the perivascular channel consisting of endothelial cells (ECs) and fibroblastic reticular cells (FRCs).** **(A)** Schematic depiction of high endothelial venules (HEVs) and the stepwise transmigration process of T- and B cells across an HEV. **(B)** Representative image of an HEV; endothelial cells (red), perivascular sheath consisting of FRCs (blue), and transmigrating T cells (green). Twenty T-cell tracks are shown. **(C)** Representative image sequence showing the stepwise migration process of T- and B cells across HEVs; adhesion to EC, intraluminal crawling, trans-EC migration, intra-PVC crawling, and trans-FRC migration. The dotted lines indicate T- and B-cell tracks. Scale bars, 10 $\mu$m. **(D, E, F, G, H, I, J)** Quantitative analysis of the migratory dynamics in the stepwise process of T- or B-cell transmigration across an HEV; time required for trans-EC and trans-FRC migration, mean velocity in the PVC and parenchyma, dwell time and path length inside the PVC. Each symbol represents a single cell. The box graph indicates the 25th and 75th percentiles; the middle line and whiskers of the box indicate the median value and standard deviation, respectively; the small square represents the mean value. The number of analysed cells is indicated below the graph. Four and three mice were used for the analysis of T- and B cells, respectively. *P*-values were calculated with the Mann–Whitney test. **(J)** Linear dependence of PVC path length on dwell time in the PVC. The solid and dotted lines represent the linear fitting of T- and B-cell data, respectively.

To further investigate the role of L-selectin/PNAd interactions on the post-luminal migration of T cells in HEVs, we used a blocking antibody against PNAds (MECA79). To induce a blocking effect on PNAds expressed on the abluminal side of HEVs while minimizing the blocking effect on PNAds expressed on the luminal side of HEVs, we injected MECA79 into a footpad instead of tail vein. Fluorescence-labelled MECA79 accumulated at high levels on the abluminal side of HEVs, whereas less accumulation was observed on the luminal side at 3 h after the injection (Fig S7). Consistent with the GlcNAc6ST-1 KO mice, MECA79 significantly increased the time required for T-cell

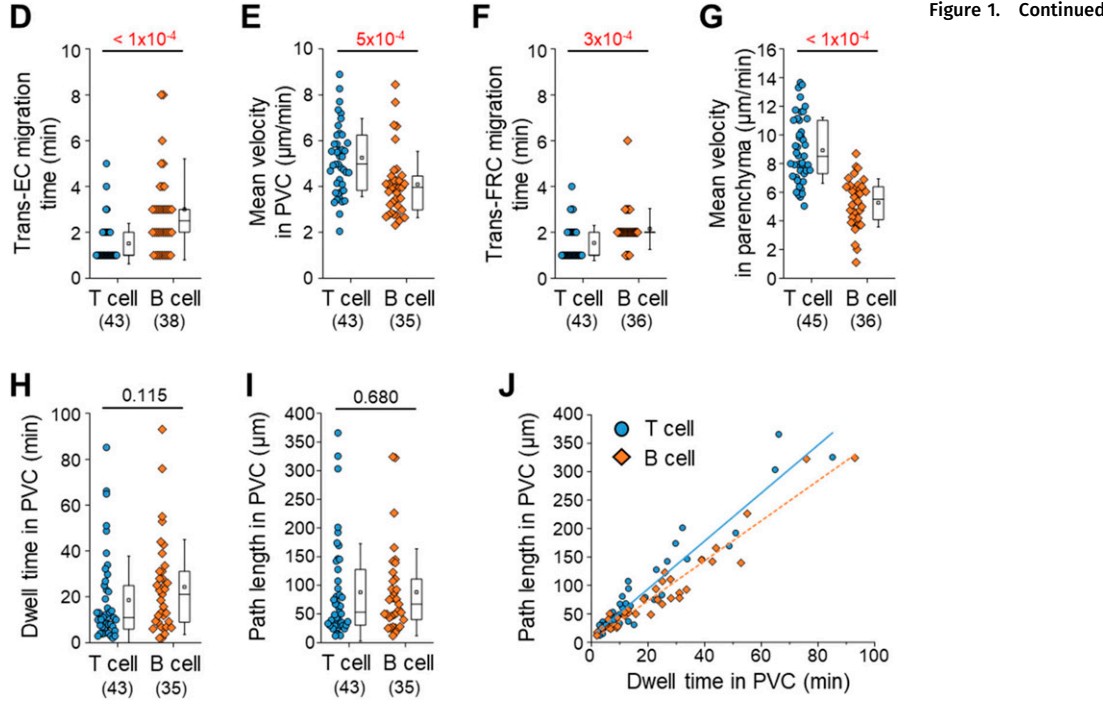

trans-FRC migration (3.1 ± 3.9 min) compared with that required with the control antibody (1.5 ± 0.8 min; Fig S8C), indicating PNAds were required for efficient T-cell trans-FRC migration. In addition, MECA79 substantially increased the trans-EC migration time (2.1 ± 0.9 min), dwell time, and path length in the PVC (32 ± 33 min, 115 ± 114 $\mu$m) and decreased the MI during intra-PVC migration (0.33 ± 0.21) compared with those in the presence of the control antibody (trans-EC migration time, 1.7 ± 0.6 min; dwell time, 13 ± 13 min; path length, 54 ± 48 $\mu$m; MI, 0.54 ± 0.43; Fig S8A, E, F, and H). The mean velocity and displacement of T cells in the PVC were similar between the two groups (Fig S8B and G). These results indicate that MECA79 delayed the passage of T cells in PVC by making the T-cell detour to an exit site. Consistent with the GlcNAc6ST-1 KO mice, MECA79 considerably decreased the mean velocity of T cells in parenchyma (6.6 ± 2.2 $\mu$m/min) compared with that induced by the control antibody (7.8 ± 2.6 $\mu$m/min; Fig S8D), indicating PNAds were also required for efficient T-cell parenchymal migration. Collectively, these GlcNAc6ST-1 KO and MECA79 experiments show that interactions between L-selectins and PNAds are also involved in the post-luminal migration of T- and B cells in HEVs from trans-EC migration to trans-FRC migration beyond their known role in luminal migration.

### T- and B cells transmigrate through the preferred sites (hot spots) in ECs and FRCs of HEVs

T cells do not transmigrate across the HEV wall in a random fashion but rather use discrete sites to arrive inside parenchyma (11). The diapedesis of T cells through discrete HEV sites has been simply described as a single step without distinction between trans-EC and trans-FRC migration (2). The aforementioned imaging method used to distinguish ECs and FRCs in HEVs enabled the observation of

trans-EC and trans-FRC migration hot spots separately. Multiple T cells sequentially transmigrated across ECs at the same site (Fig 3A and Video 3), and multiple T cells sequentially passed though FRCs at the same site to arrive inside parenchyma (Fig 3B and Video 4). In addition, we also observed a B-cell trans-EC migration hot spot (Fig 3C and Video 5) and a B-cell trans-FRC migration hot spot (Fig 3D and Video 6). A 3D distribution of the trans-EC and trans-FRC migration sites in HEVs clearly shows the trans-EC and trans-FRC migration hot spots (Fig 3E). The average number of cells trans-EC migrating at a hot spot was 2.5 ± 0.1 for T cells and 2.5 ± 0.2 for B cells during 3 h (mean ± SEM, n = 14 and 10 mice for T- and B cells, respectively). In rare cases, up to five T cells or five B cells used the same site for trans-EC migration. The average number of cells trans-FRC migrating at a hot spot was 2.8 ± 0.1 for T cells and 2.4 ± 0.2 for B cells during 3 h (mean ± SEM, n = 14 and 10 mice for T- and B cells, respectively). In rare cases, up to eight T cells or five B cells used the same site for trans-FRC migration.

The aforementioned experiments involved imaging adoptively transferred T or B cells that might compete with endogenous lymph node homing cells to transmigrate across HEVs. To observe trans-EC and trans-FRC migration hot spots for endogenous lymph node homing cells, we used Kaede transgenic mice, in which all cells express the photoconvertible fluorescent protein Kaede (19). When a 405-nm laser was irradiated onto HEVs, all cells in a field of view changed from green to red (Fig S9A), and newly appearing cells (non-photoconverted cells) in the HEV lumen were green. Therefore, we were able to observe the trans-EC and trans-FRC migration of the green non-photoconverted cells across the red photoconverted ECs in HEVs (Fig S9B–D and Video 7). Up to 12 and 8 non-photoconverted cells transmigrated across ECs and FRCs, respectively, at the same sites in HEVs during 1.5 h of imaging (Fig S9B–D). Collectively, mice of the

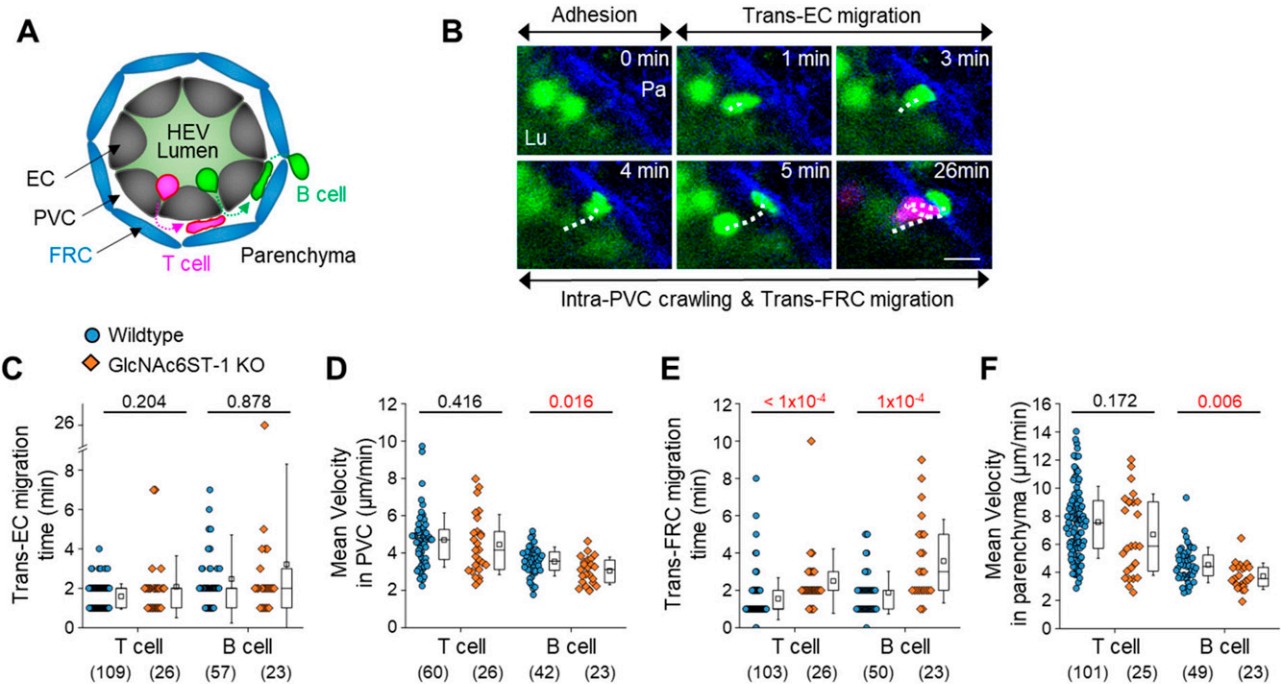

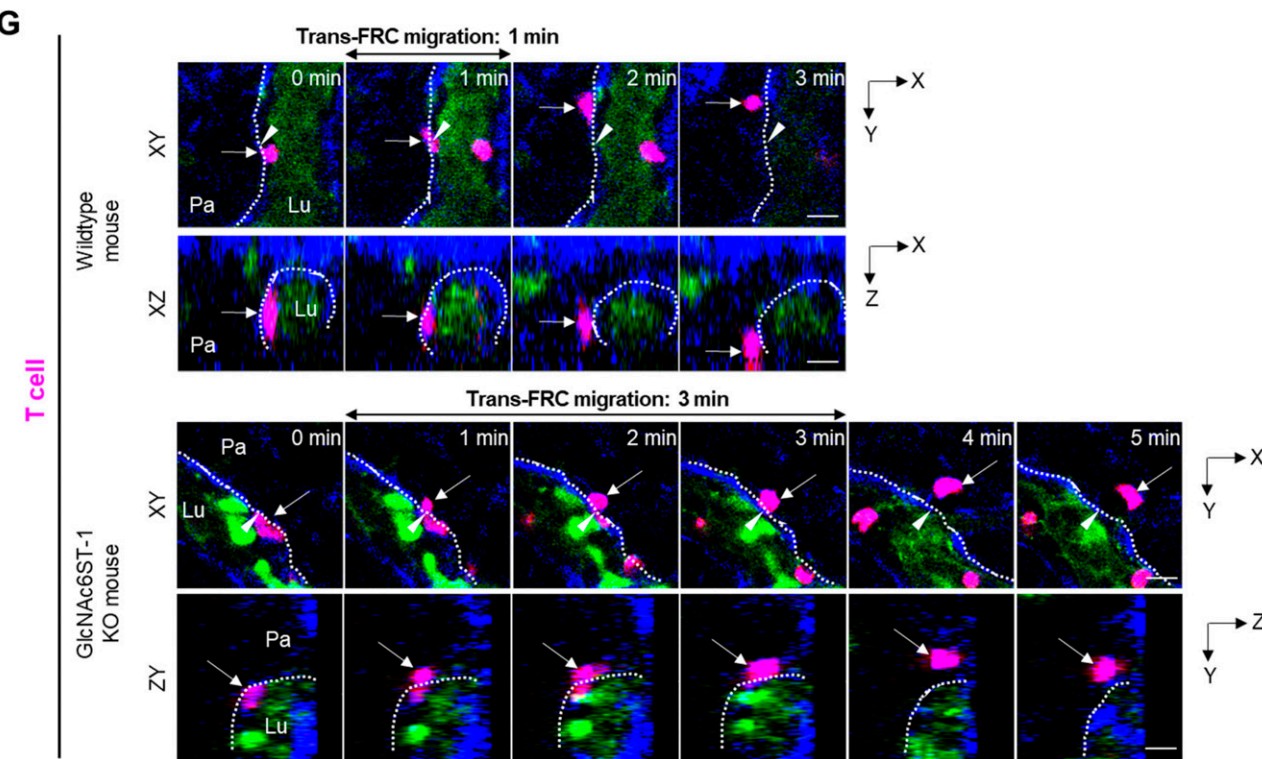

**Figure 2.    Effect of GlcNAc6ST-1 deficiency on T- and B-cell transmigration across high endothelial venules (HEVs).**
**(A)** Schematic depiction of fluorescent labelling for the simultaneous imaging of transmigrating T cells (red) and B cells (green) via a perivascular sheath consisting of fibroblastic reticular cells (blue) in GlcNAc6ST-1 KO and wild-type mice. The HEV lumen (light green) was labelled by intravenously injecting FITC-dextran, which facilitates the identification of the luminal surface in negative contrast. **(B)** Representative image sequence showing the stepwise migration process of a B cell across an HEV; adhesion to endothelial cell (EC), trans-EC migration, intra-PVC crawling and trans-fibroblastic reticular cell (FRC) migration. The dotted line indicates the B-cell track.

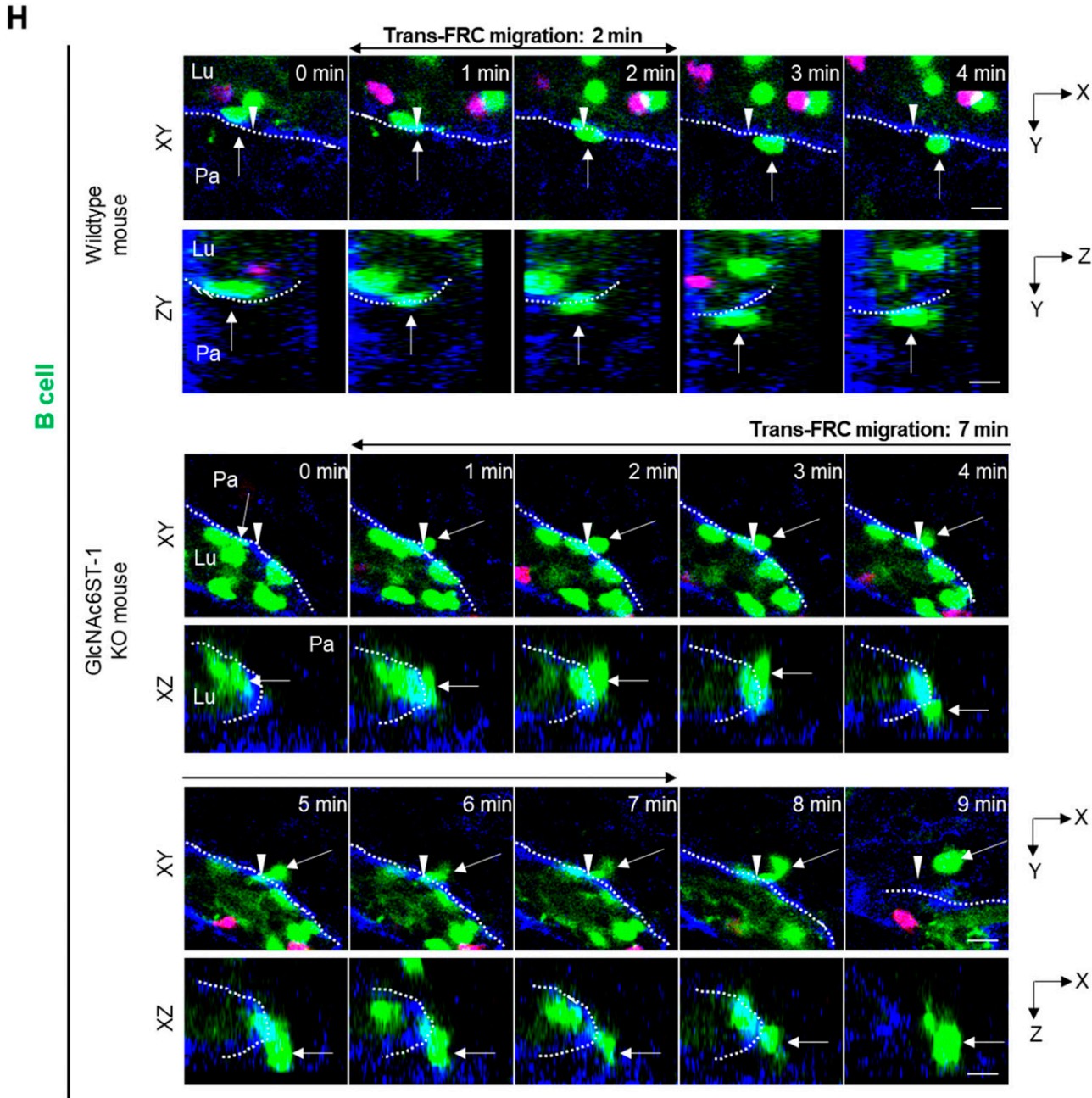

**Figure 2. Continued**

adoptive transfer model and Kaede mice clearly show the existence of trans-EC and trans-FRC T- and B-cell migration hot spots in HEVs.

Interestingly, the average number of trans-FRC migrating T cells at one site (1.8 ± 0.1, mean ± SEM, n = 14 mice) was significantly higher than that of trans-EC migrating T cells at one site (1.3 ± 0.0; Fig 3F). Furthermore, the ratio of hot spots to the total potential transmigration sites was also substantially higher for trans-FRC T-cell migration (0.42 ± 0.15, mean ± SD, n = 14 mice) than for trans-EC

Lu, lumen; Pa, parenchyma. These images correspond to a 20-μm-thick maximum intensity projection. Scale bars, 10 μm. **(C, D, E, F)** Quantitative analysis of the migratory dynamics of the stepwise T- and B-cell transmigration process across HEVs of GlcNAc6ST-1 KO mice compared with those of wild-type mice; time required for trans-EC and trans-FRC migration, mean velocity in the PVC and parenchyma. Each symbol represents a single cell. The box graph indicates the 25th and 75th percentiles; the middle line and whiskers of the box indicate the median value and standard deviation, respectively; the small square represents the mean value. The number of analysed cells is indicated below the graph. Four mice were analysed for each group. *P*-values were calculated with the Mann-Whitney test. **(C, D, E, F)** *P*-values between T- and B cells in wild-type mice were 0.0003 (C), <0.0001 (D), 0.0165 (E) and <0.0001 (F). **(G, H)** Representative image sequence showing that more time is required for trans-FRC migration in GlcNAc6ST-1 KO mice than in wild-type mice. The dotted lines indicate the boundary of FRCs. These images are serial single Z-frames (XY plane) and XZ or YZ cross sections. Arrow heads indicate the trans-FRC migration site. Scale bars, 10 μm.

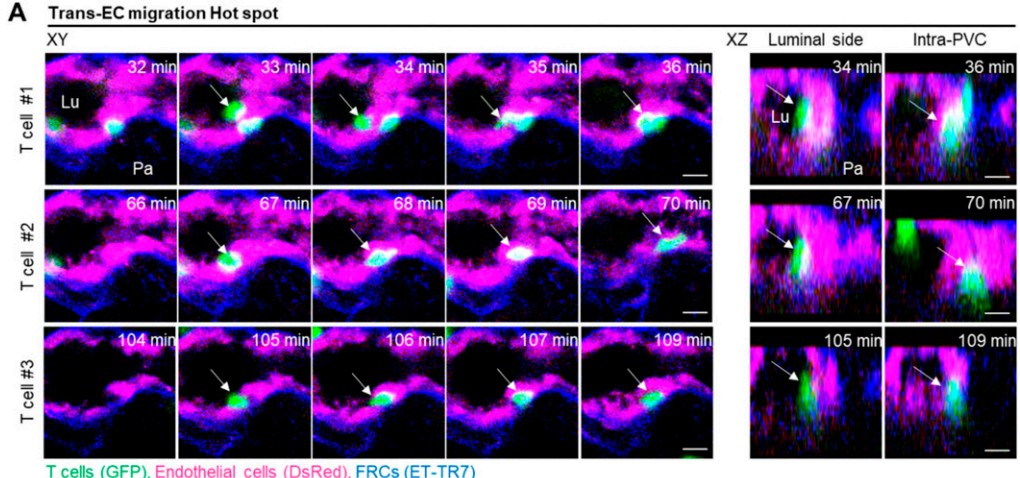

**A  Trans-EC migration Hot spot**

T cells (GFP), Endothelial cells (DsRed), FRCs (ET-TR7)

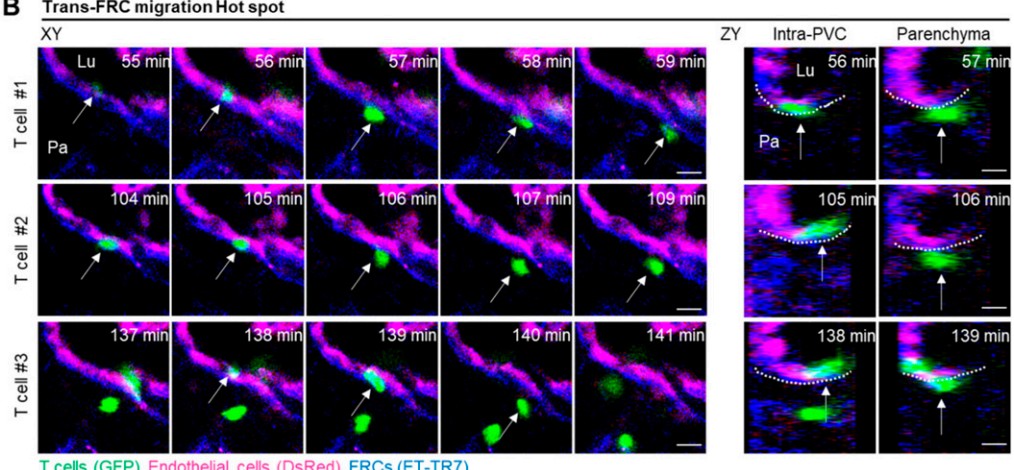

**B  Trans-FRC migration Hot spot**

T cells (GFP), Endothelial cells (DsRed), FRCs (ET-TR7)

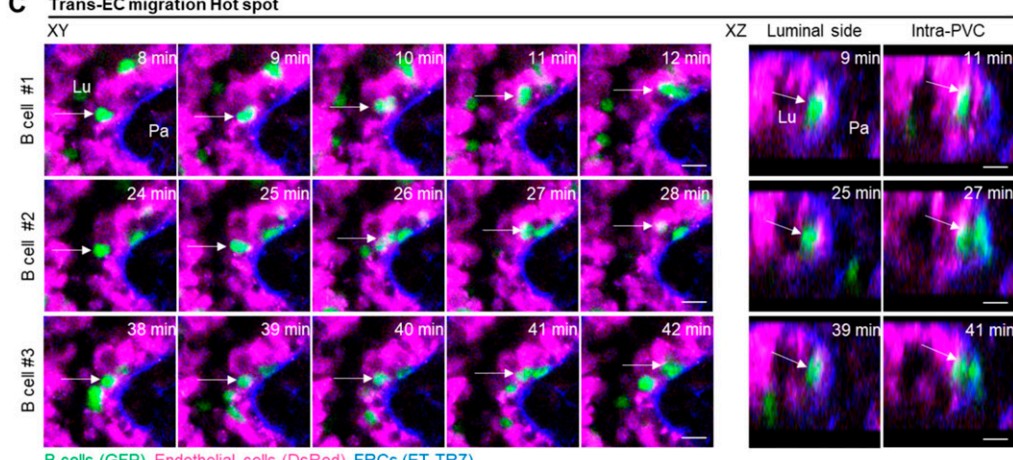

**C  Trans-EC migration Hot spot**

B cells (GFP), Endothelial cells (DsRed), FRCs (ET-TR7)

**Figure 3. Hot spots for trans-endothelial cell (EC) and trans-fibroblastic reticular cell (FRC) T- and B-cell migration.**
**(A, C)** Representative image sequence showing T- and B cells (green, arrow) transmigrate across the EC (red) at the same site, entering the PVC from the lumen. Lu, lumen; Pa, parenchyma. **(B, D)** Representative image sequence showing T- and B cells (green, arrow) transmigrate across the FRC (blue) at the same site, exiting from the PVC to parenchyma. Lu, lumen; Pa, parenchyma. Scale bars, 10 μm. **(E)** Representative 3D reconstructed image showing the distribution of trans-EC (green dots) and trans-FRC migration sites (red dots) of T- and B cells in high endothelial venules. The number of T- and B cells transmigrating at the same site is indicated. Scale bars, 10 μm. **(F)** Average numbers of labelled T- and B cells transmigrating at one site for 3 h. **(G)** Ratio of hot spots to total transmigration sites for 3 h. The hot spot is defined as a site of ECs or FRCs where more than two T or B cells transmigrate across the ECs or the FRCs in high endothelial venule. Each symbol represents a single mouse. The box graph indicates the 25th and 75th percentiles; the middle line and whiskers of the box indicate the median and standard deviation, respectively; the small square represents the mean value. Fourteen (34 ± 18 cells/mouse for trans-EC migration, 30 ± 13 cells/mouse for trans-FRC migration) and 10 mice (22 ± 15 cells/mouse for trans-EC migration, 17 ± 12 cells/mouse for trans-FRC migration) were analysed for T- and B cells, respectively. *P*-values were calculated with paired *t* tests.

T-cell migration (0.19 ± 0.09; Fig 3G). For B cells, there were no significant differences between the trans-EC and trans-FRC migration, which might have been partially due to the low number of B cells analysed per mouse (22 ± 15 cells/mouse for trans-EC migration; 17 ± 12 cells/ mouse for trans-FRC migration) compared with that for T cells (34 ± 18 cells/mouse for trans-EC migration, 30 ± 13 cells/mouse for trans-FRC migration). These results imply that trans-FRC T-cell migration is confined to fewer sites than trans-EC T-cell migration.

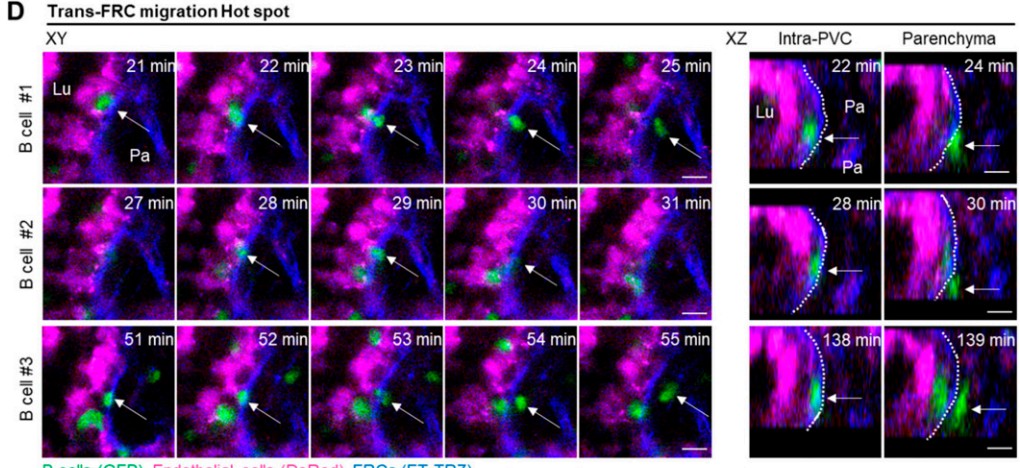

**D** Trans-FRC migration Hot spot

B cells (GFP), Endothelial cells (DsRed), FRCs (ET-TR7)

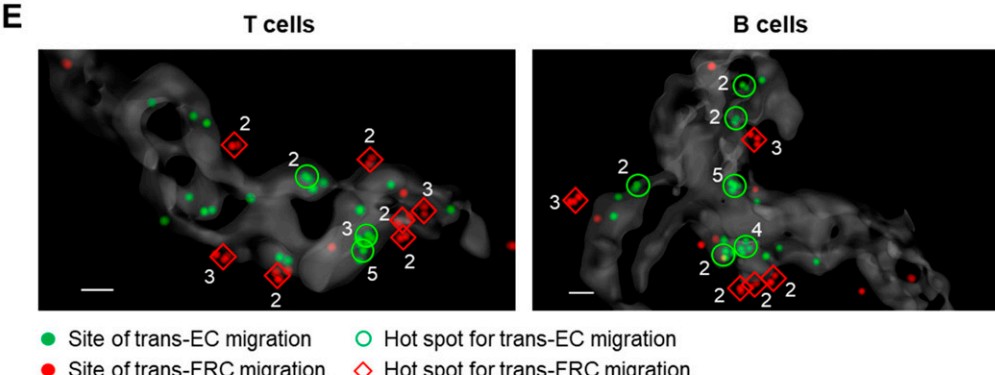

**E**

● Site of trans-EC migration  ○ Hot spot for trans-EC migration
● Site of trans-FRC migration  ◇ Hot spot for trans-FRC migration

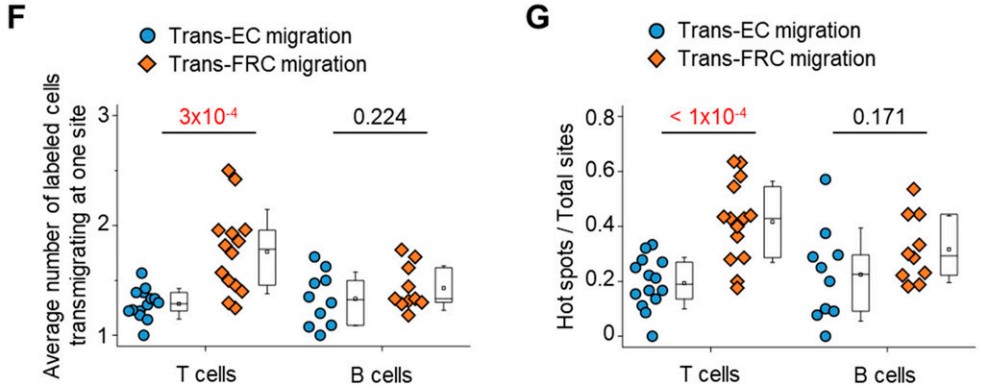

## T- and B cells preferentially share hot spots for trans-FRC migration but not for trans-EC migration

Simultaneously imaging T- and B cells showed that some T- and B cells transmigrated across FRCs at the same site (Fig 4A and Video 8). To investigate whether T- and B cells share their hot spots preferentially or accidently, we compared the percentage of T-cell hot spots in total B-cell hot spots (diamond symbols in Fig 4B) with its predicted value that is the possibility of accidently sharing

T- and B-cell hot spots (round symbols in Fig 4B). The predicted value can be calculated as the percentage of T-cell hot spots in total transmigration sites. To note, the percentage of hot spots in total sites for trans-FRC migration was higher than that for trans-EC migration (Fig 3G and round symbols in Fig 4B) maybe because the number of trans-FRC migration sites was less than that of trans-EC migration sites. It implies that the possibility of accidently sharing T- and B-cell hot spots for trans-FRC migration is higher than that for trans-EC migration. However, surprisingly, the percentage of

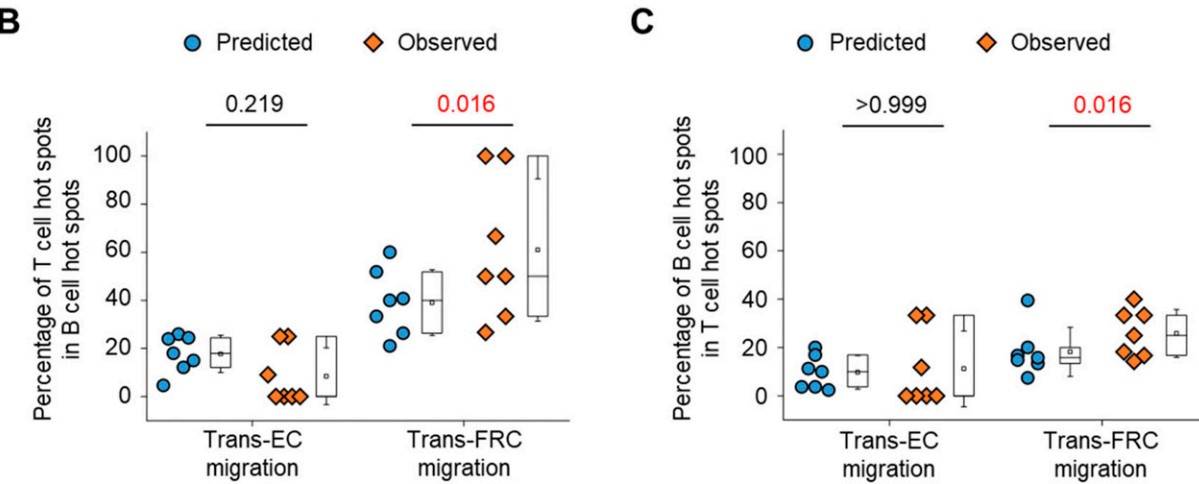

**Figure 4. T- and B cells preferentially share hot spots for trans-fibroblastic reticular cell (FRC) migration but not for trans-endothelial cell migration.**
**(A)** Representative image sequence showing T (red, arrow) and B cells (green, arrow) transmigrate across FRCs (blue) at the same site. The dotted lines indicate the boundary of FRCs. Lu, lumen; Pa, parenchyma. These images correspond to a 20-$\mu$m-thick maximum intensity projection. Scale bars, 10 $\mu$m. **(B, C)** The round and diamond symbols represent predicted and observed values, respectively, for the percentage of T-cell hot spots in B-cell hot spots (B), for the percentage of B-cell hot spots in T-cell hot spots (C). Each symbol represents a single mouse. The box graph indicates the 25th and 75th percentiles; the middle line and whiskers of the box indicate the median and standard deviation, respectively; the small square represents the mean value. Seven mice (43 ± 20 T cells and 24 ± 16 B cells/mouse for trans-endothelial cell migration, 38 ± 14 T cells and 19 ± 13 B cells/mouse for trans-FRC migration) were analysed. *P*-values were calculated with Wilcoxon test.

T-cell hot spots in B-cell hot spots was significantly higher than its predicted value of accidently sharing hot spots for trans-FRC migration (Fig 4B). Similarly, the percentage of B-cell hot spots in T-cell hot spots was also significantly higher than its predicted value for trans-FRC migration (Fig 4C). These results imply that T- and B cells preferentially share trans-FRC migration hot spots beyond the prediction for accidently sharing. However, there were no significant differences between observed and predicted values for trans-EC migration (Fig 4B and C), which implies T- and B cells just accidently share their trans-EC migration hot spots.

### T- and B cells prefer to transmigrate across FRCs covered by perivascular CD11c+ DCs

Neutrophils preferentially extravasate close to perivascular macrophages in inflamed skin vessels (16). In lymph nodes, many DCs are positioned close to HEVs (20, 21). Elimination of DCs in a lymph node impairs lymphocyte recruitment to the lymph node (22). Based on these facts, we next investigated the possible association of trans-FRC migration hot spots with perivascular DCs in HEVs. To simultaneously image DCs with T or B cells, we adoptively transferred DsRed-expressing T or B cells into a CD11c-YFP mouse (23) (Fig 5A and B). During the intravital imaging of HEVs, many T cells transmigrated across FRCs at the same site covered by perivascular CD11c+ DCs (Fig 5C and Video 9). To determine whether T cells transmigrate across FRCs covered by the CD11c+ DCs preferentially or accidently, we compared the percentage of trans-FRC migration sites covered by DCs (described as Type 1 in Fig S11) with DC coverage on the HEV (Fig S10). For T cells, the percentages of trans-FRC migration sites and hot spots covered by DCs (69% ± 10% and 78% ± 11%, respectively) were significantly higher than the DC coverage on HEVs (55% ± 12%; Fig 5D). For B cells, the percentages of trans-FRC migration sites and hot spots covered by DCs (66% ± 14%, 79% ± 34%) were considerably higher than the DC coverage on HEVs (49% ± 14%; Fig 5D). These results reveal that T- and B cells preferentially transmigrate across FRCs covered by DCs. Furthermore, additional 14% T cells and 22% B cells also contacted with surrounding DCs during trans-FRC migration although their trans-FRC migration sites were not covered by DCs (described as Type 2 or Type 3 in Fig S11). Collectively, these suggest that perivascular DCs in HEVs may regulate the trans-FRC migration of T- and B cells in HEVs.

## Discussion

The transmigration of leukocytes across blood vessel walls is a key event in host defence reactions and immune system homeostasis (2, 24). Over the past several decades, most studies have focused on the interactions between leukocytes and ECs, the first cellular barrier in the blood vessel wall (25). After trans-EC migration, leukocytes must pass pericytes, the second and final cellular barrier in the blood vessel wall. Recently, Proebstl et al clearly visualized the post-trans-EC migration of neutrophils in inflamed tissue by 3D time-lapse intravital microscopy with fluorescent labelling of ECs, pericytes and neutrophils in different colours (15). They showed that intercellular adhesion molecule 1 (ICAM-1), Mac-1, and LFA-1 mediate neutrophil crawling in the narrow

space between ECs and pericytes and that neutrophils prefer to exit through the enlarged pericyte gap in inflamed tissue (15).

Unlike inflamed blood vessels, HEVs constantly recruit lymphocytes into lymph nodes in the steady state and therefore have different cellular and molecular characteristics (2, 17). HEVs are composed of cuboidal ECs and pericyte-like FRCs. Herein, we observed the post-luminal migration of T- and B cells, including trans-EC, intra-PVC and trans-FRC migration, in HEVs by fluorescently labelling ECs, FRCs and T or B cells different colours. We uncovered that PNAd expressed on the abluminal side of HEVs are involved in the post-luminal migration of T- and B cells and that T- and B cells prefer to transmigrate through FRCs covered by CD11c+ DCs.

PNAds expressed in HEV ECs mediate lymphocyte rolling and sticking. Although PNAds are also expressed at the endothelial junction and on the abluminal side of HEVs, their involvement in post-luminal lymphocyte migration has not been investigated as extensively as that in luminal migration because experimental methods such as molecular-deficient mice and blocking antibody treatment cause severe defects in luminal migration. In contrast, the significant defects in post-luminal leukocyte migration could be observed in inflamed cremaster venules of L-selectin–deficient mice because of no defect in the luminal migration (26). To selectively block the function of molecules expressed on the abluminal side of HEVs, we herein used GlcNAc6ST-1–deficient mice (8, 9, 10) or injected blocking antibodies via the footpad rather than via intravenous injection, as previously reported (15). GlcNAc6ST-1 is predominantly involved in PNAd expression on the abluminal side rather than on the luminal side, although GlcNAc6ST-1 deficiency also modestly affects the luminal migration of lymphocytes by increasing the rolling velocity (9). GlcNAc6ST-1 deficiency increased the time required for trans-FRC migration but not that for trans-EC migration. This could be attributable to deficiency of GlcNAc6ST-1–synthesizing L-selectin ligands in the abluminal side of HEV. In addition to the abluminal side of HEV ECs, FRCs also express GlcNAc6ST-1, but not GlcNAc6ST-2 (27), implying that FRC-expressed GlcNAc6ST-1 may regulate trans-FRC migration in some extent. We also investigated the effect of MECA79 on abluminal migration because GlcNAc6ST-1 deficiency does not eliminate all PNAd expression on the abluminal side of HEVs. The blocking antibody MECA79 increased the time required for trans-FRC migration, the dwell time and path length in the PVC and decreased the MI in the PVC. Thus, PNAds expressed at the endothelial junction and on the abluminal side of HEVs facilitate the efficient transmigration of lymphocytes across the HEV wall but do not slow transmigration in the perivascular region. GlcNAc6ST-1 deficiency and MECA79 antibody also decreased the parenchymal B- and T-cell velocities immediately after extravasation, respectively, probably because of blockade of parenchymal expression of PNAd in close proximity to HEV (6, 21, 28). A caveat is that all antibodies we used contain a preservative, sodium azide, that has potential side effects on lymphocyte migration in lymph node (29). Nevertheless, Fig S3 shows no significant difference in T-cell migration in HEV between anti-ER-TR7–injected and noninjected groups.

Interestingly, blocking of L-selectin shedding also slows down the post-luminal lymphocyte migration (30, 31). This report combined with our result may imply that an appropriate amount of L-selectin expression on lymphocyte is important for the efficient abluminal migration. What is

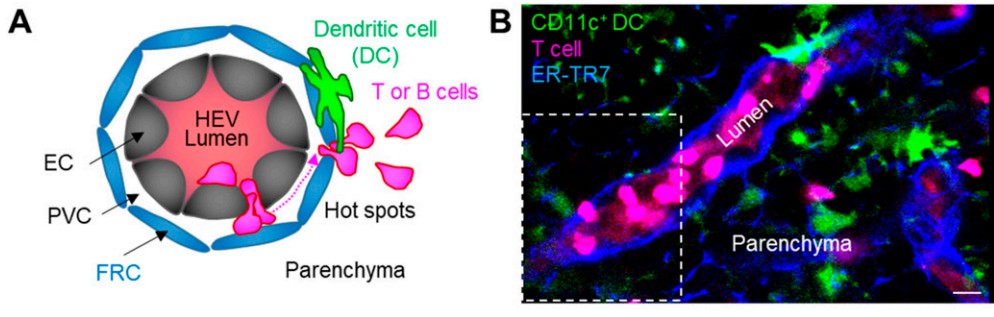

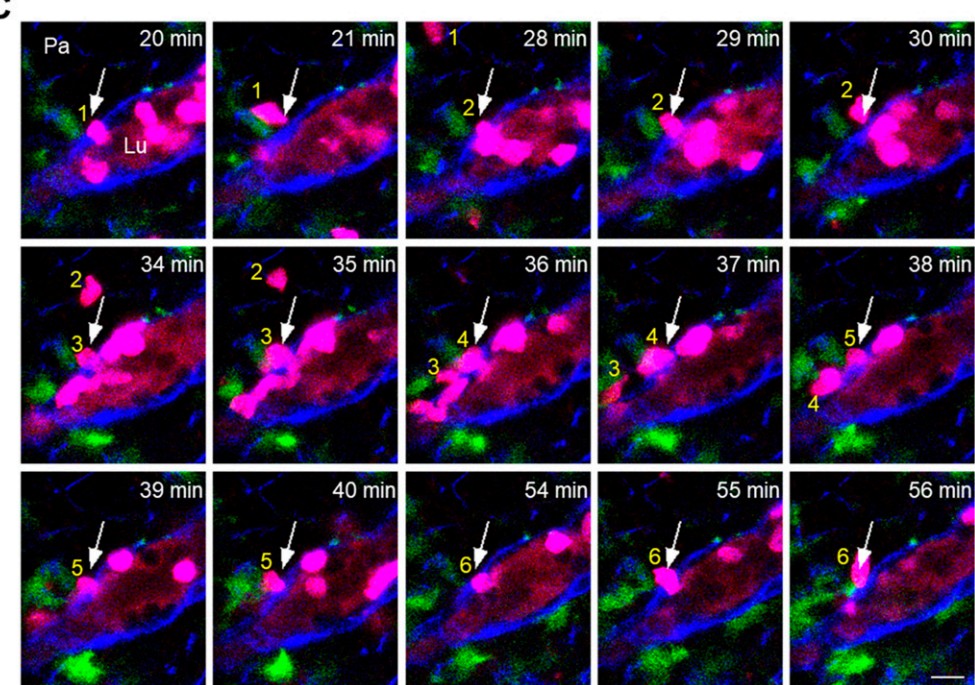

**Figure 5. T- and B cells transmigrate across fibroblastic reticular cells (FRCs) covered by CD11c+ DCs.**
**(A)** Schematic depiction of fluorescent labelling for simultaneously imaging DCs (green), T or B cells (red) and FRCs (blue). The high endothelial venule (HEV) lumen (light red) was labelled by intravenously injecting TRITC-dextran, which facilitates the identification of the luminal surface in negative contrast. **(B)** Representative image of an HEV with a CD11c+ DC and a T-cell trans-FRC migrating from the PVC to parenchyma. Scale bar, 10 μm. **(C)** Representative image sequence showing that six T cells transmigrate across the FRC at the same site (arrow) in close proximity to CD11c+ DCs. These images correspond to a 6-μm-thick maximum intensity projection. Scale bars, 10 μm. **(D)** Comparison of the coverage of CD11c+ DCs on HEVs and trans-FRC migration sites or hot spots covered by DCs. The hot spot is defined as sites of FRCs where more than two T or B cells transmigrates across the FRCs in HEV. Each symbol represents a single mouse. The box graph indicates the 25th and 75th percentiles; the middle line and whiskers of the box indicate the median and standard deviation, respectively; the small square represents the mean value. Eight (48 ± 18 T cells/mouse) and eight mice (21 ± 19 B cells/mouse) were used for the analysis of T- and B cells, respectively. *P*-values were calculated with paired *t* test.

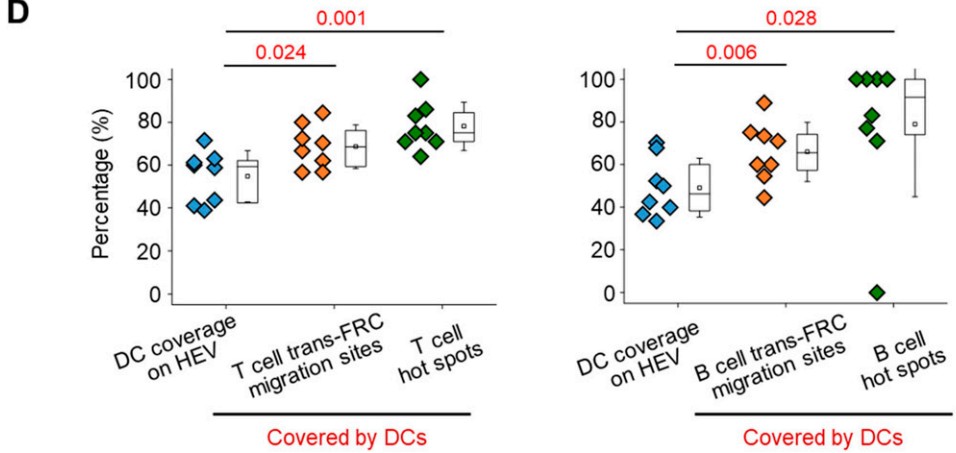

not yet clear is the exact timing of lymphocyte L-selectin shedding, although that of monocytes occurs during trans-EC migration (32).

The molecular mechanism underlying the PNAd-regulated lymphocyte migration in abluminal side of HEV where shear stress would be absent may be related to the L-selectin signalling of lymphocytes triggered by interaction with PNAds. L-selectin signalling activates β2 integrins on lymphocytes (33, 34) and enhances the chemotaxis of lymphocytes to CCL21 (35). Interestingly, knocking out both

GlcNAc6ST-1 and GlcNAc6ST-2 completely eliminates PNAd expression in HEVs, but the other L-selectin ligands remain on the abluminal side of HEVs (8). Recently, the reported antibodies CL40 and S2 were shown to react with more types of L-selectin ligands than MECA79, which reacts with only PNAds (2, 36, 37). These antibodies or L-selectin chimeric proteins (38) may be used to induce a stronger blocking effect on abluminal L-selectin ligands than that induced by MECA79.

In addition to PNAds, important molecules have been suggested to play a role in the abluminal migration of lymphocytes in HEVs, such as CCL21, autotaxin and mac25/angiomodulin. CCL21 secreted by ECs and FRCs binds to collagen IV on the abluminal side of HEVs (39). CCL21 activates LFA-1 integrins of lymphocytes by interacting with CCR7 (1). Autotaxin secreted by the ECs of HEVs produces lysophosphatidic acid, which facilitates lymphocyte release from the HEV endothelium to parenchyma (40). Mac25/angiomodulin localized exclusively to the abluminal side of HEVs interacts with chemokines, such as CCL21, but its exact contribution to lymphocyte migration is unclear (41).

FRCs form a reticular conduit network in lymph nodes by secreting and surrounding collagen fibres (42, 43). The FRC conduit delivers small molecules, such as antigens, from afferent lymph to the HEV lumen (1, 42, 43). Using this function, we visualized FRCs surrounding HEVs in a popliteal lymph node by injecting a fluorescence-labelled anti–ER-TR7 antibody into a mouse footpad. ER-TR7 antigens are ECM components secreted by FRCs (44) and form the conduit structure ensheathed by FRCs (11). Although ER-TR7 is a well-known FRC marker (43), the function of the ER-TR7 antigen is unknown. Recently, one study showed that administration of an ER-TR7 antibody into tolerant mice affects the HEV basement membrane structure and CCL21 distribution (45). Although our analysis showed no difference in T-cell transmigration across HEVs between the ER-TR7 antibody–injected and noninjected groups in the steady state, the increase in the parenchymal T-cell velocity of the antibody-injected group (Fig S3) implied the contribution of the ER-TR7 antigen to T-cell migration in the lymph node parenchyma and required that the antibody be used with caution. FRCs can be identified by their expression of ER-TR7, podoplanin, $\alpha$-SMA, and CCL19, which are not expressed in HEV ECs (43). Thus, CCL19-cre;loxP-EGFP mice (46) or $\alpha$-SMA-GFP mice (15, 47) can be used for the intravital fluorescent imaging of HEV FRCs without requiring the ER-TR7 antibody footpad injection.

Resident DCs in lymph nodes strategically position on the FRC conduit, including HEVs, to monitor the antigens delivered from afferent lymph (42). These DCs contribute to lymphocyte entry into lymph nodes by interacting with HEV ECs and FRCs. CD11c+ DCs maintain the HEV EC phenotype, including the expression of GLYCAM1 (L-selectin ligands), by lymphotoxin-$\beta$-receptor (LT$\beta$R)-dependent signalling (22). FRCs also express LT$\beta$R, and its signalling affects FRC expansion in inflamed lymph nodes (48), but the effects of LT$\beta$R signalling triggered by DCs on lymphocyte entry into lymph nodes in the steady state are unknown. Interestingly, podoplanin expressed in FRCs regulates HEV endothelial adherens junctions by interacting with the CLEC-2 of platelets (49). LT$\beta$R or podoplanin signalling may be related to the hot spots of trans-EC lymphocyte migration.

The podoplanin of FRCs also controls FRC contractility (50, 51) and ECM production (52) by interacting with the CLEC-2 of DCs in inflamed lymph nodes. In the steady state, resident DCs in lymph nodes express CLEC-2 (53). Thus, it is conceivable that CLEC-2+

resident DCs may control the contractility of FRCs and remodel ECM surrounding HEVs to facilitate the trans-FRC migration of T- and B cells. Thus, the CLEC-2/podoplanin signalling may represent a key molecular mechanism underlying our discovery that trans-FRC migration hot spots preferentially occur at FRCs covered by CD11c+ DCs. In addition, the PVC (a narrow space between ECs and FRCs) acts as a region of waiting for entering lymphocytes to maintain their population in lymph nodes when lymphocyte egress is blocked (54), which may also be regulated by the DC-FRC interaction in HEVs.

Although we observed T- and B cells preferentially transmigrate across FRCs covered by CD11c+ DCs (Fig 5), there is a possibility of existence of independent mechanisms with no causal relationship that facilitates the selected trans-FRC sites permits lymphocyte diapedesis and at the same time favors DC accumulation. To exclude this possibility, additional experiments such as observing the change of hot spots after elimination of perivascular DCs followed by finding and validating the role related molecules are required. In addition, better characterization of the CD11c+ DCs located in the hot spots of HEVs is required to differentiate them from the other CD11c+ DCs observed in the non–hot spot regions of HEVs. Some T-cell-zone resident macrophages can also express CD11c (55). Imaging of a triple-transgenic mouse with Zbtb46-cre;tdTomato and CD11b-GFP will be able to differentiate three types of DCs and macrophages potentially associated with the hot spots: Zbtb46$^+$CD11b$^-$ cDC1 (red), Zbtb46$^+$CD11b$^+$ cDC2 (yellow), and Zbtb46$^-$CD11b$^+$ macrophage (green) (55, 56).

Herein, we clearly visualized the hot spots of trans-EC T- and B-cell migration in HEVs in vivo, but we unfortunately did not elucidate their underlying mechanisms. According to a previous report on neutrophils in inflamed cremaster muscle venules, neutrophils preferentially adhere to the endothelial junction region (13) and preferentially transmigrate across the ECs through the junction (para-cellular route) rather than through the nonjunctional site (trans-cellular route) (12). In lymph node HEVs, distinguishing between para- and trans-cellular routes has been challenging for several reasons. HEV ECs (diameter, 20–30 $\mu$m) (57) are plump in shape and smaller than flat ECs of the cremaster muscle venules (major axis, 47 $\mu$m; minor axis, 23 $\mu$m) (58). The average time required for the trans-EC migration of T cells in HEVs (<2 min) is lower than that required for neutrophils in inflamed venules (6 min) (12). Although our previous report showed the possibility of visualizing in vivo paracellular T-cell migration in HEVs by high-speed (30 frames/s) confocal microscopy and fluorescently labelling the EC surface with an anti-CD31 antibody (59), the preferential route of trans-EC lymphocyte migration remains unknown.

Compared to T cells, B cells took a longer time to pass EC and FRC layers in HEV and had lower velocity in PVC and parenchyma just after extravasation. Furthermore, the adhesion rate of B cells to HEV EC in luminal side is lower than that of T cells (5). These could be attributed to lower expression of L-selectin and CCR7 on B cells than T cells (18, 60). The difference in homing efficiency between T- and B cells may vary depending on the HEV location because of the heterogeneous expression of chemokines and integrins on HEV EC and surrounding FRCs in peripheral lymph node (27, 61). The HEVs imaged in this work were located around 40–70 $\mu$m depth from the capsule where might be close to B-cell follicles. B-cell homing efficiency in the deeper paracortical T-cell zone could be different from our data probably due to less CXCL13 that is chemoattractant

for B cells highly expressed in follicles. Although this work focused on peripheral lymph node, the other lymphoid organs have different lymphocyte homing efficiency (62) because of organ-specific gene expression on HEVs (63). B cells home better to mesenteric lymph nodes and Peyer's patches than peripheral lymph nodes (62) by CD22-binding glycans expressed preferentially on the HEVs of mesenteric lymph nodes and Peyer's patches (63).

Inflamed peripheral lymph nodes become larger by recruiting more lymphocytes and even L-selectin–negative leukocytes that are excluded in the steady state (64, 65). Inflamed HEV ECs show different gene expression, such as down-regulation of GLYCAM1 and GlcNAc6ST-1 (61). In addition, inflamed HEV integrity may be loosened because of markedly increased leukocyte influx although the HEV FRCs can prevent bleeding by interacting with platelet CLEC-2 (49). CD11c+ DCs are associated with inflamed HEV EC proliferation that is functionally associated with increased leukocyte entry (66). The stepwise migration of lymphocyte across inflamed HEVs and their hot spots with peri-vascular CD11+ DCs will be interesting topic for future study.

In conclusion, we clearly visualized and analysed the multiple steps involved in post-luminal T- and B-cell migration, including trans-EC, intra-PVC, and trans-FRC migration, in HEVs, suggesting that these migration steps are regulated by PNAds. Notably, we identified the trans-EC and trans-FRC migration hot spots separately in HEVs. Our analysis revealed that T- and B cells preferentially share their trans-FRC migration hot spots but not trans-EC migration hot spots. In addition, the trans-FRC migration of T cells was confined to fewer sites than trans-EC migration. Surprisingly, the trans-FRC migration of T- and B cells prefer-entially occurred at FRCs covered by CD11c+ DCs. These results imply that pericyte-like FRCs, the second cellular barrier of HEVs, regulate the entry of T- and B cells to maintain peripheral lymph node homeostasis more precisely and restrictively than we previously thought.

# Materials and Methods

### Mice

Actin-DsRed and actin-GFP mice were kindly provided by Dr Gou Young Koh (KAIST, Daejeon, Republic of Korea). GlcNAc6ST-1 KO mice were previously described (9, 10). Kaede (19) and CD11c-YFP (23) mice were generously provided by Dr Michio Tomura (Kyoto University, Kyoto, Japan) and Dr Jae-Hoon Choi (Hanyang University, Seoul, Republic of Korea), respectively. C57BL/6 mice were pur-chased from the Jackson laboratory. All mice were maintained on a C57BL/6 background and bred in our SPF facility at KAIST. 8–16-wk-old mice were used. Experiments were approved by the Animal Care Committee of KAIST (KA2013-11).

### In vivo fluorescent labelling

T or B cells ($2–4 × 10^7$) obtained from two spleens of actin-GFP mice by negative MACS kits (114.13D; Thermo Fisher Scientific, MAGM204; R&D Systems) were intravenously injected to an actin-DsRed mouse. Higher than 95% purity of isolated T- and B cells was confirmed by

FACS analysis using pan-T-cell and pan-B-cell markers, CD3e and B220, respectively. HEV ECs of actin-DsRed mouse popliteal lymph node expressed red fluorescence much stronger than the surrounding stromal cells and endogenous lymphocytes, which was suffi-cient to image only HEV ECs by adjusting an image contrast (Fig 1A and B). To fluorescently visualize FRCs surrounding HEVs of a popliteal lymph node, an anit–ER-TR7 antibody conjugated with Alexa Fluor 647 (10 $\mu$g, 50 $\mu$l, sc-73355 AF647; Santa Cruz) was injected into a footpad 12 h before imaging. For the footpad injection of the antibody, we anesthetized a mouse by intra-peritoneal injection of a mixture of 10 mg/kg Zoletil (Virbac) and 6 mg/kg xylazine.

For simultaneous imaging of T- and B cells, T- and B cells isolated from spleens of actin-DsRed and actin-GFP mice, respectively, were intravenously injected to a wild-type mouse. To fluorescently label HEV lumen, FITC-dextran (2 MD, 0.2 mg/ml, 50 $\mu$l, FD2000S; Sigma-Aldrich) dissolved in 1× PBS (Lonza) was intravenously injected, which facilitated the identification of EC in negative contrast (Fig 2A and B). To distinguish B cells (bright green) from the HEV lumen (light green; Fig 2A and B), a low concentration of FITC-dextran in blood was maintained by the intravenous injection of a small amount of FITC-dextran repeatedly with a tail vein catheter during imaging. For simultaneous imaging of CD11c+ DCs and T or B cells, T or B cells obtained from two spleens of actin-DsRed mice were intravenously injected to a CD11c-YFP mouse. To fluorescently label HEV lumen, TRITC-dextran (500 kD, 1 mg/ml, 100 $\mu$l, 52194; Sigma-Aldrich) dissolved in 1× PBS (Lonza) was intravenously injected.

### Blocking antibody

To test accumulation of blocking antibody in abluminal side of HEV, we used MECA79 or immunoglobulin M (IgM) control conjugated with Alexa Fluor 488 (10 $\mu$g, 20 $\mu$l, 53-6036-82, 53-4341-80; eBio-science). For blockade of PNAd, MECA79 or IgM control (25 $\mu$g, 50 $\mu$l, 553863, 553940; BD Biosciences) were injected into a footpad about 3 h before imaging.

### Flow cytometry

Popliteal and inguinal lymph nodes were harvested and single-cell suspensions were prepared by mechanical dissociation on a cell strainer (RPMI-1640 with 10% FBS). Cell suspensions were centri-fuged at 300$g$ for 5 min. Erythrocytes in lymph nodes were lysed with ammonium-chloride-potassium lysis buffer for 5 min at RT. Cell suspensions were washed and filtered through 40-$\mu$m filters. Nonspecific staining was reduced by using Fc receptor block (anti-CD16/CD32). Cells were incubated for 30 min with varying com-binations of the following fluorophore-conjugated monoclonal antibodies: anti-CD3e (clone 145-2C11; BD pharmigen), anti-CD4 (clone GK1.5; BD Pharmingen), anti-CD8 (clone 53-6.7; eBioscience), anti-CD44 (clone IM7; BioLegend), and anti-CD62L (clone MEL-14; eBioscience) antibodies (diluted at a ratio of 1:200) in FACS buffer (5% bovine serum in PBS). After several washes, cells were ana-lysed by FACS Canto II (BD Biosciences) and the acquired data were further evaluated by using FlowJo software (Treestar).

## Mouse preparations and intravital imaging

Mouse was anesthetized by intraperitoneal injection of a mixture of 20 mg/kg Zoletil (Virbac) and 11 mg/kg xylazine. Depth of anesthesia was continuously monitored during the experiment by using a toe pinch and maintained by additional intramuscular injection of half dose of the initially injected Zoletil-xylazine mixture whenever a response was observed. The left popliteal lymph node of the anesthetized mouse was surgically exposed by small incision of skin and fascia at popliteal fossa. In addition, fatty tissue covering the lymph node was carefully removed by micro-dissection forceps. During the entire intravital imaging, core body temperature of the mouse was maintained at 36°C by using a temperature regulating system consisting of heating pad and rectal probe (Kent Scientific Corp.). Temperature of the surgically exposed lymph node was maintained at 36–38°C by using tissue temperature sensor, and warm water recirculator (Kent Scientific Corp.) or silicone rubber heater (NISSI-YGC). We catheterized tail vein to inject T or B cells immediately before the imaging, and to inject FITC-dextran or TRITC-dextran repeatedly during the imaging. The prepared mouse on the motorized animal stage is shown in Fig S12. For 3D-time-lapse imaging, 22 sequential z-stacks (170 × 170 $\mu$m, 512 × 512 pixels) with a 2-$\mu$m axial spacing were acquired at intervals of a minute for 2–3 h after injection of lymphocytes.

## Confocal microscopy system

Intravital imaging was performed by using a custom-built laser scanning confocal microscope (59, 67). Three continuous-wave lasers with 488 nm (Cobolt, MLD), 561 nm (Cobolt, Jive) and 640 nm (Cobolt, MLD) were used as excitation lights for fluorescence imaging. Fluorescence signals were simultaneously detected by three bandpass filters (FF01-525/50, FF01-600/37, FF01-685/40; Semrock) and three photomultiplier tubes (R9110; Hamamatsu). For photoconversion of Kaede proteins, HEV in a field of view (170 × 170 $\mu$m) was irradiated by 405 nm laser (~10 mW/mm$^2$; Coherent; OBIS) for 5 min. Z-axis resolution of about 3 $\mu$m per section was acquired with 100 $\mu$m pinhole and 60× objective lens (LUMFLN, water immersion, NA 1.1; Olympus).

## Image processing and data analysis

Tracking T or B cells was performed by using IMARIS software (version 8.1.2; Bitplane) or manual tracking plugin of ImageJ (National Institutes of Health). The position error of the track generated from tissue drift was corrected by drift correction function of IMARIS, or by a custom-written MATLAB program for the x-y axis and manually selection of same plane for z axis. We manually distinguished the multistep of lymphocyte transmigrations including trans-EC, intra-PVC, trans-FRC, and intra-parenchyma migrations, and calculated the various parameters such as the mean velocity and the required time for each step by using Excel (Microsoft Corp.) and MATLAB (MathWorks). The 3D wind-rose plot of intra-PVC migrations and the 3D distribution of trans-EC and trans-FRC migration sites in HEV were made by MATLAB and IMARIS software, respectively. We defined a hot spot as a site where two or more T or B cells

transmigrated during 2–3 h of imaging. The DC coverage on HEV (Fig 5D) was the average of two measurements of DC coverage at start (0 min) and end time (180 min) of imaging. The DC coverage on HEV was calculated by dividing FRC volume colocalized with DC (cyan) by total FRC volume of HEV (blue; Fig S10). The FRC volume colocalized with DC was measured by surface–surface colocalization function of IMARIS. Some DCs covering HEVs were moving for 3 h imaging. We observed that some T or B cells transmigrated across a hot spot with temporarily absent of DC although the hot spot was covered by DC for the other T- or B-cell trans-FRC migration. For counting the hot spots covered by DCs in Fig 5D, we included the hot spots that were covered by DC for at least half of T- or B-cell trans-FRC migration.

## Statistics

Mann–Whitney test was used for the comparison of T- and B cells, GlcNAc6ST-1 KO and wild-type, MECA79 and control antibody, anit–ER-TR7 antibody-injected group and noninjected group. Paired $t$ test was used for the comparison of trans-EC and trans-FRC migration (Fig 3F and G), DC coverage and trans-FRC migration sites covered by DCs (Fig 5D). Wilcoxon test was used for the comparison of prediction and observation of transmigration sites where T- and B cells simultaneously exit (Fig 4C). One-way ANOVA Tukey's test was conducted to analyze the change in parenchymal T-cell velocity over time. $P < 0.05$ was considered statistically significant.

# Supplementary Information

# Acknowledgements

We thank Dr Sukhyun Song, Dr Jin Sung Park and Dr Kangsan Kim for help with FACS analysis, and Dr Gou Young Koh, Soyeon Ahn, Jingu Lee, Dr Inwon Park, Dr Yoonha Hwang, Dr Howon Seo, Ryul Kim, Dr Jinhyo Ahn, Eunji Kong, Sujung Hong for helpful discussion. This work was supported by the Global Frontier Project (NRF-2013M3A6A4044716), and the Basic Research Program (NRF-2017R1E1A1A01074190) of National Research Foundation of Korea (NRF) through funded by the Ministry of Science and ICT, Republic of Korea.

## Author Contributions

K Choe: conceptualization, resources, formal analysis, investigation, visualization, methodology, and writing—original draft, review, and editing.
J Moon: resources, formal analysis, and investigation.
SY Lee: resources.
E Song: resources.
JH Back: resources.
J-H Song: methodology.
Y-M Hyun: writing—review and editing.
K Uchimura: resources and writing—review and editing.

P Kim: conceptualization, supervision, and writing—original draft, review, and editing.

## Conflict of Interest Statement

The authors declare that they have no conflict of interest.

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
