## [Reviewer comments · Life Science Alliance]

Life Science Alliance

Stepwise transmigration of T and B cells through a perivascular channel in high endothelial venules

Kibaek Choe, Jieun Moon, Soo Lee, Eunjoo Song, Ju Back, Joo-Hye Song, Young-Min Hyun, Kenji Uchimura, and Pilhan Kim

DOI: <https://doi.org/10.26508/lsa.202101086>

Corresponding author(s): Pilhan Kim, Korea Advanced Institute of Science and Technology

Review Timeline:

Submission Date:	2021-04-06
Editorial Decision:	2021-05-27
Revision Received:	2021-06-03
Accepted:	2021-06-09

Transaction Report:

Please note that the manuscript was reviewed at Review Commons and these reports were taken into account in the decision-making process at Life Science Alliance.

May 27, 2021

RE: Life Science Alliance Manuscript #LSA-2021-01086-T

Prof. Pilhan Kim
Korea Advanced Institute of Science and Technology
Graduate School of Nanoscience and Technology, Korea Advanced Institute of Science and
Technology
291 Daehak-ro
Daejeon, - 305-701
Korea, Republic of

Dear Dr. Kim,

Thank you for submitting your revised manuscript entitled "Stepwise transmigration cascade of T and B cells through the perivascular channel in lymph node high endothelial venules". We would be happy to publish your paper in Life Science Alliance pending text revisions to acknowledge Reviewer 3's remaining concern about the possible interference of Sodium Azide in lymphocyte migration, and final revisions necessary to meet our formatting guidelines.

- please add a Category and a Running Title for your manuscript in our system
- please add a Summary Blurb/Alternate Abstract in our system
- please upload your main and supplementary figures as single files
- please upload your main manuscript text as an editable doc file;
- please add Author Contributions to our system, as well
- please rename the "Competing interests" section to "Conflict of Interest"
- please add your main, supplementary figure, and video legends to the main manuscript text after the references section
- please add callouts for Figures 3A; S1A, B; S2A-C; S3B-F; S5A; S6A-C; S9D; S10A, B; movie S1 to your main manuscript text
- please revise the inset position in Figure 5C so that they match the zoomed-in parts
- please add scale bars in Figure S10B

A. FINAL FILES:

B. MANUSCRIPT ORGANIZATION AND FORMATTING:

Sincerely,

Shachi Bhatt, Ph.D.
Executive Editor

Reviewer #1 (Comments to the Authors (Required)):

This is an interesting, well executed study that makes a significant contribution to understanding lymphocyte recruitment and entry into lymph nodes. I am happy with the Authors responses and amendments to the paper. I recommend publication without further changes.

Reviewer #2 (Comments to the Authors (Required)):

The authors have adequately addressed my questions. I have no further comments.

Reviewer #3 (Comments to the Authors (Required)):

Unfortunately, authors failed to reply adequately to the point I raised, which is the following. The only antibody blocking experiments they performed to obtain mechanistic insights was by the use of commercially available monoclonal antibodies, all of which unfortunately contained a preservative, sodium azide, which potently blocks lymphocyte migration in vivo (Freitas AA & Bognacki J, Immunol 36:247, 1979). Therefore, the results of these antibody blocking experiments cannot be taken at face value.

June 9, 2021

RE: Life Science Alliance Manuscript #LSA-2021-01086-TR

Prof. Pilhan Kim
Korea Advanced Institute of Science and Technology
Graduate School of Nanoscience and Technology, Korea Advanced Institute of Science and
Technology
291 Daehak-ro
Daejeon, - 305-701
Korea, Republic of

Dear Dr. Kim,

Thank you for submitting your Research Article entitled "Stepwise transmigration of T and B cells through a perivascular channel in high endothelial venules". It is a pleasure to let you know that your manuscript is now accepted for publication in Life Science Alliance. Congratulations on this interesting work.

DISTRIBUTION OF MATERIALS:

Again, congratulations on a very nice paper. I hope you found the review process to be constructive and are pleased with how the manuscript was handled editorially. We look forward to future exciting

submissions from your lab.

Sincerely,
